# Spectral Rank Certification for Foundation Model Adapters

Mohammed AHNOUCH
Université Paris 1
Paris, France

Lotfi Elaachak
FSTT Tangier
Tangier, Morocco

## Abstract

Low-rank adapter pruning often thresholds singular values without a calibrated false-positive rate. We study a finite-dimensional reference experiment: an $m \times n$ Gaussian matrix either contains only noise or contains a rank-one spike with unknown Haar-uniform directions. The main result is an exact chi-square divergence series with coefficients given by even spherical moments. This gives a computable Le Cam certificate at the dimensions of a concrete layer, not only in an asymptotic regime. The same formula recovers the rectangular Baik–Ben Arous–Peche (BBP) subcritical limit, and a ratio bound certifies numerical truncation. We also give a compact-manifold Laplace expansion showing that the mixture likelihood is not a pure threshold in $s_1$: finite-sample evidence contains the gap factor $s_1^{|m-n|} \prod_{i \geq 2}(s_1^2 - s_i^2)$. The contribution is therefore a reference-model calibration layer for adapter spectra, with explicit asymptotic limits and stated conditions for empirical-null deployment.

## CCS Concepts

• **Mathematics of computing** → *Distribution functions*; *Probabilistic algorithms*.

## Keywords

low-rank adaptation, adapter pruning, spiked random matrices, finite-sample calibration, spectral testing

## 1 Motivation and Scope

Low-rank adaptation (LoRA) replaces a dense fine-tuning update by a low-rank adapter [5]. Adaptive variants such as AdaLoRA, DoRA, VB-LoRA, and LoRA-Mini allocate or parameterize rank more carefully [7–9, 12]. A complementary post-training question is statistical: for an observed adapter spectrum, which singular components are distinguishable from a layer-specific null?

We address this question through a reference model rather than a universal model of trained adapters. The null and alternative are

$$H_0 : Y = \sigma G, \qquad H_1 : Y = \theta uv^\top + \sigma G, \qquad (1)$$

where $G_{ij} \overset{\text{iid}}{\sim} N(0, 1)$ and $u \sim \text{unif}(S^{m-1})$, $v \sim \text{unif}(S^{n-1})$ are independent. The Haar prior encodes rotational ignorance about the spike orientation rather than a distributional model of trained adapters: it is the unique rotation-invariant prior on $(u, v)$ and is therefore least favorable for a spike of unknown direction. Its Bayes risk is a valid lower bound for the fixed-but-unknown direction problem, because the minimax power over directions is at most the prior-average power; a selector that is weak on average against this prior is consequently weak against its worst-case direction.

The paper makes four contributions. First, the second moment between the null and the Haar mixture is computed exactly at fixed $(m, n)$, yielding a Le Cam certificate rather than an asymptotic one.

Second, this series recovers the rectangular Baik–Ben Arous–Peche (BBP) phase transition—the asymptotic singular-value transition for spiked random matrices [1, 2]—as its high-dimensional limit. Third, the integrated likelihood is shown to depend on the full spectrum, not on $s_1$ alone. Fourth, the calibrated-diagnostic claims are delimited exactly: the reference model certifies detectability and calibration, whereas rank selection for a deployed adapter additionally requires an empirical null, downstream metrics, and adaptive-rank baselines.

## 2 Exact Finite-Sample Certificate

For a selector $\phi(Y) \in \{0, 1\}$ let $\alpha(\phi) = \mathbb{P}_0\{\phi = 1\}$ and $\pi(\phi; \theta) = \mathbb{P}_1\{\phi = 1\}$, where $P_1$ is the Haar mixture in (1). The mixture likelihood ratio is

$$L(Y) = \mathbb{E}_{u,v} \exp \left\{ \frac{\theta}{\sigma^2} \langle Y, uv^\top \rangle - \frac{\theta^2}{2\sigma^2} \right\}. \qquad (2)$$

**Theorem 1 (Exact chi-square divergence).** *Let $(u, v)$ and $(u', v')$ be independent draws from $\text{unif}(S^{m-1}) \times \text{unif}(S^{n-1})$ and set $\lambda = \theta^2/\sigma^2$. Then*

$$\chi^2(P_1 \| P_0) = \mathbb{E} \exp\{\lambda \langle u, u' \rangle \langle v, v' \rangle\} - 1$$

$$= \sum_{j=1}^{\infty} \frac{\lambda^{2j}}{(2j)!} M_m(2j) M_n(2j), \qquad (3)$$

*where, for independent $x, x' \sim \text{unif}(S^{d-1})$,*

$$M_d(2j) = \mathbb{E}\langle x, x' \rangle^{2j} = \frac{(2j - 1)!!}{d(d + 2) \cdots (d + 2j - 2)}.$$

**Proof.** Write $A = uv^\top$ and $A' = u'v'^\top$. Under $P_0$, conditional on $A, A'$, the Gaussian moment-generating function gives

$$\mathbb{E}_0 L(Y)^2 = \mathbb{E}_{A,A'} \exp \left\{ -\theta^2/\sigma^2 + \frac{\theta^2}{2\sigma^2} \|A + A'\|_F^2 \right\}.$$

Because $\|A + A'\|_F^2 = 2 + 2\langle u, u' \rangle \langle v, v' \rangle$, the first identity follows. Expanding the exponential is justified by boundedness. Odd terms vanish by symmetry, and independence separates the even moments. Rotational invariance fixes $x' = e_1$; the first coordinate of $x \sim \text{unif}(S^{d-1})$ has beta density proportional to $(1 - t^2)^{(d-3)/2}$, giving the stated $M_d(2j)$. □

The series is entire at fixed $m, n$ but it also has a certified finite computation. Let $a_j$ be the $j$th term in (3), $S_K = \sum_{j \leq K} a_j$, and $R_K = \chi^2 - S_K$. For $q_j = \lambda^2/[(m + 2j)(n + 2j)]$, if $q_{K+1} < 1$ then

$$0 \leq R_K \leq \frac{a_{K+1}}{1 - q_{K+1}}. \qquad (4)$$

Indeed, $a_{j+1}/a_j = \lambda^2(2j + 1)/[(2j + 2)(m + 2j)(n + 2j)] \leq q_j$, and $q_j$ decreases. The crude global bound $R_K \leq e^{|\lambda|}\mathbb{P}\{\text{Poisson}(|\lambda|) \geq 2K + 2\}$ is also valid since $M_m(2j)M_n(2j) \leq 1$.

COROLLARY 2 (BBP-SCALE LIMIT). *If $m, n \to \infty$ and $\lambda^2/(mn) \to \beta \in [0, 1)$, then*

$$\chi^2(P_1\|P_0) \to (1 - \beta)^{-1/2} - 1.$$

*Equivalently, if $\theta/[\sigma(mn)^{1/4}] \to c < 1$, the limit is $(1 - c^4)^{-1/2} - 1$.*

For fixed $j$, $M_m(2j)M_n(2j) = ((2j-1)!!)^2(mn)^{-j}\{1 + o(1)\}$, so the $j$th term converges to $\binom{2j}{j}(\beta/4)^j$. A geometric domination from (4) justifies summing, and the central-binomial generating function gives the limit.

The statistical consequence is the finite-sample Le Cam bound

$$\pi(\phi; \theta) \le \alpha(\phi) + \frac{1}{2}\sqrt{\chi^2(P_1\|P_0)}. \tag{5}$$

It follows from $\mathbb{E}_1\phi - \mathbb{E}_0\phi \le \mathrm{TV}(P_0, P_1) \le \frac{1}{2}\sqrt{\chi^2(P_1\|P_0)}$, where the last step is Cauchy–Schwarz applied to $L - 1$ [6, 10]. Since the left side is prior-average power, (5) also upper-bounds the worst-direction power of any fixed-size selector by the same prior-average argument.

## 3 Spectral Rules versus Likelihood

The unnormalized null edge is approximately $\sigma(\sqrt{m}+\sqrt{n})$, while the additive rectangular outlier scale is $\sigma(mn)^{1/4}$ [2]. For $m = n = d$, define $\widetilde{Y} = Y/(\sigma\sqrt{d})$ and suppose $\theta/(\sigma\sqrt{d}) \to c$. The rank-one rectangular outlier theorem gives

$$s_1(\widetilde{Y}) \to \begin{cases} 2, & 0 \le c \le 1, \\ c + c^{-1}, & c > 1, \end{cases} \tag{6}$$

in probability. Thus a fixed asymptotic edge is a scale, not a finite-sample false-positive rate. A calibrated spectral rule should use the null quantile

$$q_{1-\alpha}^{(0)} = \inf\{t : \mathbb{P}_0(s_1(Y) \le t) \ge 1-\alpha\}, \quad \phi_\alpha(Y) = \mathbf{1}\{s_1(Y) > q_{1-\alpha}^{(0)}\}.$$

The likelihood ratio (2) is not a function of $s_1$ alone. Let

$$I_\kappa(Y) = \int_{S^{m-1}} \int_{S^{n-1}} e^{\kappa u^\top Y v} \, d\bar\omega_m(u) d\bar\omega_n(v),$$

where $d\bar\omega_d$ is normalized surface measure. If $s_1 > s_2$ and $r = \min(m, n)$, the Morse-Laplace theorem on compact manifolds [11, Ch. VIII] gives

$$\log I_\kappa(Y) = \kappa s_1 - \frac{m+n-2}{2}\log\kappa + C(Y) + o(1),$$

$$C(Y) = \log 2 + \frac{m+n-2}{2}\log(2\pi) - \log(\omega_m\omega_n)$$

$$- \frac{1}{2}\log\left[s_1^{|m-n|}\prod_{i=2}^{r}(s_1^2 - s_i^2)\right]. \tag{7}$$

The two maximizers are $(u_1, v_1)$ and $(-u_1, -v_1)$. In tangent coordinates around $(u_1, v_1)$,

$$u^\top Y v = s_1 - \frac{s_1}{2}(\|x\|^2 + \|y\|^2) + \sum_{i=2}^{r} s_i x_i y_i + O(\|(x, y)\|^3),$$

so $\det(-H) = s_1^{|m-n|}\prod_{i=2}^{r}(s_1^2 - s_i^2)$. Hence $I_\kappa$ and $s_1$ agree only at leading exponential order. Two matrices with the same $s_1$ can receive different integrated-likelihood evidence when their leading gaps differ. Near $s_1 = s_2$, the nondegenerate expansion fails and a block statistic is more appropriate.

**Table 1: Synthetic operating points, shown as empirical FPR/power.**

| $\theta$ | $s_1$ cal. | edge | edge+2$\sigma$ | energy cal. | fixed $\tau = 25$ |
|---|---|---|---|---|---|
| 8.5 | .050/.089 | .129/.215 | .000/.000 | .050/.061 | .000/.000 |
| 12 | .050/.314 | .129/.492 | .000/.000 | .050/.233 | .000/.000 |
| 18 | .050/1.00 | .129/1.00 | .000/.716 | .050/.999 | .000/.531 |
| 30 | .050/1.00 | .129/1.00 | .000/1.00 | .050/1.00 | .000/1.00 |

For a rank-$r$ signal $X_\star = U\Theta V^\top$ with independent Haar frames and $\Theta = \mathrm{diag}(\theta_1, \ldots, \theta_r)$,

$$\chi_r^2 + 1 = \mathbb{E}\exp\{\sigma^{-2}\mathrm{tr}(\Theta U^\top U'\Theta V'^\top V)\}.$$

If $Z = \mathrm{tr}(\Theta U^\top U'\Theta V'^\top V)$, then $\mathbb{E}Z = 0$, odd moments vanish by sign symmetry, and

$$\mathbb{E}Z^2 = \frac{\|\Theta\|_F^4}{mn}, \qquad \chi_r^2 = \frac{\|\Theta\|_F^4}{2\sigma^4 mn} + \mathcal{R}_4, \tag{8}$$

with $|\mathcal{R}_4| \le \sum_{k\ge 4}(\|\Theta\|_F^2/\sigma^2)^k/k!$. Thus componentwise diagnostics ignore the leading cross term $2\theta_i^2\theta_j^2/(2\sigma^4 mn)$; clustered near-critical components should be tested as a block or handled by a global rank budget.

## 4 Diagnostic Workflow and Use Cases

The reference calculation becomes useful only after specifying a null. In a real adapter study, the null can come from multiple fine-tuning seeds, residual-tail fitting, sign flips, permutations, or task-preserving bootstraps; the assumptions of the chosen null should be reported. A layer-wise procedure is then: compute the SVD of $\widehat{\Delta}_\ell$, estimate or sample the null for the statistic, convert singular components to empirical $p$-values, and choose ranks by a fixed level, Benjamini–Hochberg false-discovery control [3], or a latency-driven rank budget. After data-dependent deflation the residual is conditioned on the extracted subspace and is no longer distributed as $\sigma G$; the Gaussian reference therefore applies rigorously only to the first component, and an empirical residual null or null-only simulation must replace it at each subsequent stage.

The numerical implementation uses log-domain summation of (3) and selects $K$ until the dimension-aware bound (4) is below the requested tolerance. Null quantiles and powers below use independent random streams, exact dense singular values, and Wilson intervals for reported false-positive rates. Table 1 gives a compact synthetic sanity check for $m = n = 128$, $\sigma = 1$, and spike strengths (30, 18, 12, 8.5) using 5000 independent null draws and 2000 independent alternative draws per spike. Entries are empirical false-positive rate/power; the calibrated $s_1$ threshold targets $\alpha = 0.05$, and Wilson intervals for the asymptotic edge false-positive rate give 0.129 with interval [0.120, 0.139]. On synthetic data with a known null the table isolates three effects the reference model is built to expose: the asymptotic edge runs at a finite-sample level of 0.129 rather than the nominal 0.05, the near-critical spikes $\theta \in \{8.5, 12\}$ have calibrated power well below one, and the supercritical spikes $\theta \in \{18, 30\}$ are detected almost surely. Establishing the same effects on trained adapters requires an empirical null.

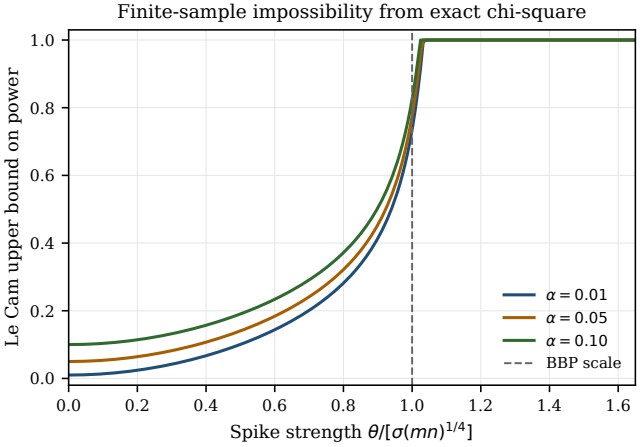
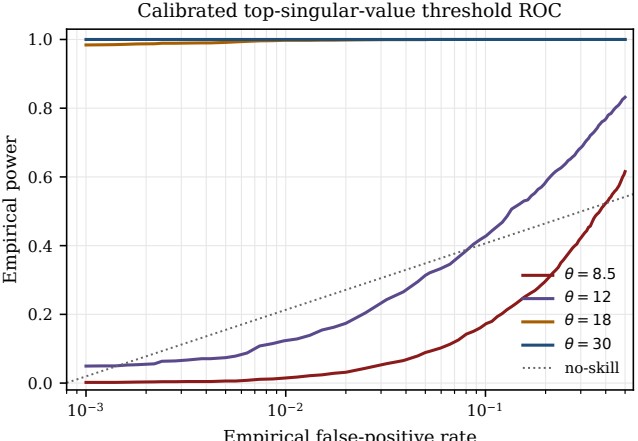

**Figure 1: Left: Le Cam power upper bounds from the exact chi-square series at $m = n = 128$. Right: calibrated top-singular-value ROC curves from independent null and alternative simulations.**

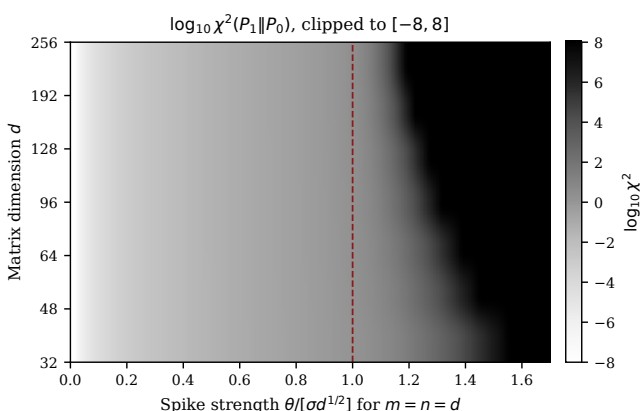
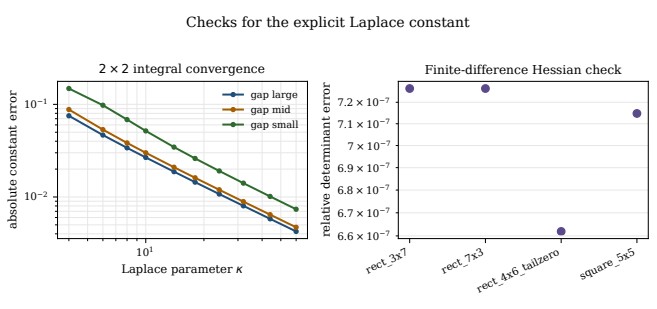

**Figure 2: Left: exact finite-sample $\log_{10} \chi^2(P_1\|P_0)$ across square dimensions, clipped to $[-8, 8]$. Right: numerical checks of the Laplace constant and Hessian determinant in (7).**

A second diagnostic applies the simulated nulls sequentially to a rank-four signal. Table 2 reports the nominal residual dimensions $(m - j + 1, n - j + 1)$ used at each stage and Benjamini–Hochberg retention at $q = 0.10$. The retention frequencies are exact under the stated sequential-null model; they do not by themselves certify that data-dependent deflation preserves a Gaussian residual, which is exactly why each stage should draw its null from held-out seeds or task-preserving randomizations on real adapters. The table fixes the quantities such a procedure must report: component index, null dimension, empirical $p$-value, and selection frequency.

Reference-model calibration also suggests stress tests for future real-adapter work. A layer should be audited under at least one non-iid null if its residual spectrum is anisotropic, heavy-tailed, or structured by the transformer block. A robust report should therefore include: the null construction, the statistic being calibrated, independent calibration/evaluation splits, uncertainty intervals for false-positive rates, retained ranks, and downstream

**Table 2: Componentwise rank-four diagnostic; BH retention at $q = 0.10$.**

| $j$ | $\theta_j$ | residual size | median $p_j$ | retention |
|---|---|---|---|---|
| 1 | 30.0 | $128 \times 128$ | .0014 | 1.000 |
| 2 | 18.0 | $127 \times 127$ | .0014 | 1.000 |
| 3 | 12.0 | $126 \times 126$ | .164 | .351 |
| 4 | 8.5 | $125 \times 125$ | .758 | .020 |
| mean retained rank | – | – | – | 2.37 |

accuracy/latency tradeoffs. These requirements lie outside the reference model but are necessary conditions for a deployable pruning claim.

The scope of the method is precise. It certifies whether a component is statistically surprising under a stated null, places several

## Operating points for a synthetic low-rank profile

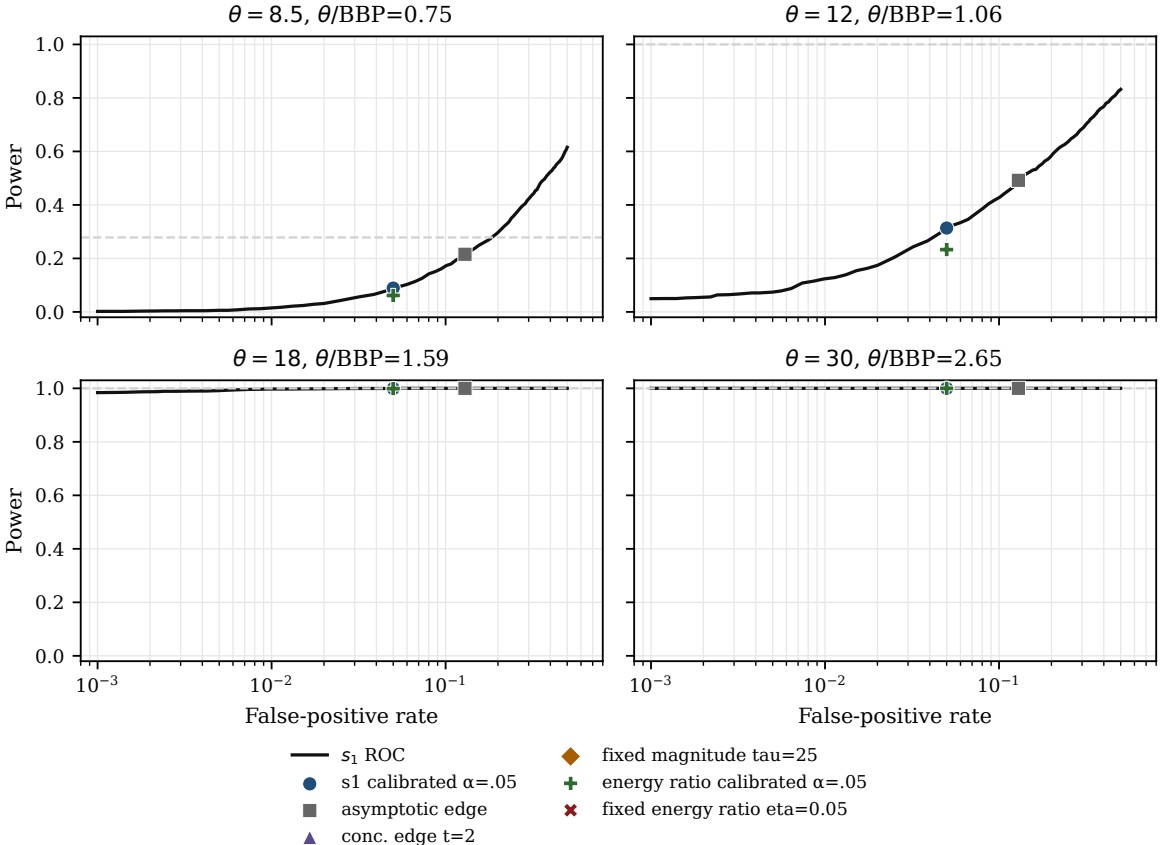

**Figure 3: Operating points for the same synthetic profile. The panels emphasize that calibration changes the reported false-positive rate, while strong spikes remain detectable.**

scalar rules on a common false-positive axis, and identifies coupled components that require a joint test. It does not select the task-optimal rank: that decision requires pruning the adapter and measuring parameter, memory, latency, and accuracy effects. Separating the statistical evidence from the engineering tradeoff lets each be evaluated on its own terms.

The resulting use cases are concrete. First, the series and (5) quantify when low-FPR retention is information-theoretically impossible under an unknown-direction reference, in the spirit of detection limits for spiked rectangular models [4]. Second, calibrated ROCs place edge, energy, and likelihood-inspired rules on a common false-positive axis. Third, (8) identifies when a layer should be pruned by a block or a budgeted rank rather than by independent singular-value tests. Downstream claims should still report retained ranks, accuracy, memory or latency, and seed variability against AdaLoRA/DoRA/VB-LoRA and energy-pruning baselines. The reference model supplies a calibrated statistical-evidence label per component; it complements rather than replaces that benchmark.

**Table 3: What the diagnostic certifies.**

| Question | Reported evidence |
|---|---|
| Can any low-FPR rule work? | Le Cam bound from (3) and (5). |
| Is a threshold calibrated? | Empirical null FPR for the chosen statistic. |
| Are components coupled? | Rank-$r$ term (8); use block tests near clusters. |
| Does pruning help? | Downstream accuracy, memory/latency, and seed variability. |

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
